# Operationalizing Social Environments in Cognitive Aging and Dementia Research: A Scoping Review

**DOI:** 10.3390/ijerph18137166

**Published:** 2021-07-04

**Authors:** Rachel L. Peterson, Kristen M. George, Duyen Tran, Pallavi Malladi, Paola Gilsanz, Amy J. H. Kind, Rachel A. Whitmer, Lilah M. Besser, Oanh L. Meyer

**Affiliations:** 1Department of Neurology, University of California Davis, Sacramento, CA 95817, USA; krmgeorge@ucdavis.edu (K.M.G.); olmeyer@ucdavis.edu (O.L.M.); 2Department of Psychology, University of California Davis, Davis, CA 95616, USA; phdtran@ucdavis.edu; 3Department of Physiology and Membrane Biology, University of California Davis, Davis, CA 95616, USA; pmalladi@ucdavis.edu; 4Kaiser Permanente Northern California Division of Research, Oakland, CA 94612, USA; Paola.Gilsanz@kp.org; 5Center for Health Disparities Research, University of Wisconsin School of Medicine and Public Health, Madison, WI 53726, USA; ajk@medicine.wisc.edu; 6Health Services and Care Research Program, Department of Medicine, University of Wisconsin School of Medicine and Public Health, Madison, WI 53726, USA; 7Department of Medicine, Division of Geriatrics and Gerontology, University of Wisconsin School of Medicine and Public Health, Madison, WI 53726, USA; 8Geriatrics Research Education and Clinical Center, Department of Veterans Affairs, Madison, WI 53726, USA; 9Public Health Sciences, Division of Epidemiology, University of California Davis, Davis, CA 95616, USA; rawhitmer@ucdavis.edu; 10Alzheimer’s Disease Research Center, University of California Davis, Sacramento, CA 95817, USA; 11Department of Urban and Regional Planning, Florida Atlantic University, Boca Raton, FL 33431, USA; lbesser@fau.edu

**Keywords:** social ecological model, social context

## Abstract

Background: Social environments are a contributing determinant of health and disparities. This scoping review details how social environments have been operationalized in observational studies of cognitive aging and dementia. Methods: A systematic search in PubMed and Web of Science identified studies of social environment exposures and late-life cognition/dementia outcomes. Data were extracted on (1) study design; (2) population; (3) social environment(s); (4) cognitive outcome(s); (5) analytic approach; and (6) theorized causal pathways. Studies were organized using a 3-tiered social ecological model at interpersonal, community, or policy levels. Results: Of 7802 non-duplicated articles, 123 studies met inclusion criteria. Eighty-four studies were longitudinal (range 1–28 years) and 16 examined time-varying social environments. When sorted into social ecological levels, 91 studies examined the interpersonal level; 37 examined the community/neighborhood level; 3 examined policy level social environments; and 7 studies examined more than one level. Conclusions: Most studies of social environments and cognitive aging and dementia examined interpersonal factors measured at a single point in time. Few assessed time-varying social environmental factors or considered multiple social ecological levels. Future studies can help clarify opportunities for intervention by delineating if, when, and how social environments shape late-life cognitive aging and dementia outcomes.

## 1. Background

Research into the relationship between social environments and health promises important insights for understanding population trends and disparities [1]. The social environment is strongly correlated and interacts with the physical environment to create our broader human ecology, but the two constructs are distinct in terms of operationalization [2]. Physical environments are assessed through measurements of the immediate physical spaces in which people spend their lives (neighborhood greenspace; air quality). By contrast, operationalizing and understanding the impact of the social environment on health is more nuanced. Yen and Syme (1999) define the social environment as “the groups to which we belong, the neighborhoods in which we live, the organization of our workplaces, and the policies we create to order our lives” [2]. In other words, social environments are not physically or geographically contained. Rather, they are formed by various layers of interpersonal relationships, the cultural and demographic characteristics of our communities and societal power dynamics. These layers social policies, opportunities, resources, and norms that ultimately impact the distribution of health and disease [3]. This idea is commonly reflected in the social ecological model (Figure 1A), which acknowledges the multiple layers of one’s social environment and how interactions between them are critical in shaping—and understanding—population health outcomes and disparities [4,5].

The study of social environments may be particularly poignant for reducing the risk of cognitive aging and dementia. Alzheimer’s disease, the most common form of dementia, is currently the 6th leading cause of death in the U.S. and is expected to become more common as the population ages [6]. Studies have documented substantial racial/ethnic [7] and socioeconomic [8,9] disparities in dementia rates. Additionally, key modifiable risk factors for dementia (e.g., cardiometabolic health [10]) fall on the causal pathways by which social environments are broadly theorized to influence health outcomes (e.g., social and material resources; health behaviors; psychosocial stress [3]). Because health disparities are, by definition, rooted in social inequities (as opposed to biological differences) [11], investigating the role of social environments across the lifecourse will further our understanding of how and when to intervene.

The aim of this scoping review is to better understand how social environments have been studied in relation to cognitive aging and dementia in observational studies. We sought to document how social environments have been operationalized, the working theories for how social environments contribute to cognitive aging and dementia, and to identify knowledge gaps and opportunities for future scientific investigation.

## 2. Methods

Scoping reviews are an important tool for evidence synthesis in public health. Like systematic reviews, scoping reviews require a systematic search and prespecified approach to study selection [12]. Unlike systematic reviews, which aim to define evidence for a narrow question that often pertains to efficacy for a specific treatment or intervention, scoping reviews have much broader aims [12,13]. Specifically, scoping reviews allow researchers to summarize a field of research, identify knowledge gaps, determine the value of undertaking a systematic review in a given area, clarify concepts, or investigate research conduct [12,13]. The rationale for conducting this scoping review is to clarify the concept of social environments through investigating the question: How have social environments been operationalized in observational studies of cognitive aging and dementia? We followed the framework for scoping reviews recommended by Arksey and O’Malley (2005) and enhanced by Levac, et al., (2010) [12,14], as detailed below.

### 2.1. Search Criteria

We conducted a systematic search to identify observational studies that investigated one or more measures of the social environment as an exposure for cognitive aging and dementia. Searches were developed and performed in collaboration with a medical sciences librarian in October 2020 using MeSH and keyword terms in PubMed and Web of Science. Using MeSH terms can simplify search syntax by automatically incorporating all items indexed in the corresponding subheadings (e.g., dementia [MeSH] includes all indexed dementia subheadings for various pathologies and sub-types of dementia). Including keyword terms reduces the potential bias of subjectivity in indexing and captures recent publications not yet indexed. All dates of publication through October 2020 were included in the search, as were all geographic locations and race/ethnic populations. Search criteria emphasized longitudinal study designs and were limited to English language peer-reviewed original research and review studies. Search term syntax is provided in Table 1.

### 2.2. Inclusion/Exclusion Criteria

All identified studies were independently screened for inclusion based on title and abstract by two individuals, PM and DT. Discrepancies were reviewed and determined by consensus among the original two reviewers and two additional reviewers, KG and RP. Studies met inclusion criteria if they: (1) were an observational study or review of observational studies; (2) included one or more cognitive/dementia outcomes assessed in midlife or late life (e.g., cognitive function assessed at one or more time points, cognitive impairment, and/or dementia diagnosis); and (3) examined at least one social environment measure as an exposure of interest. We used the definition provided by Yen and Syme (1999), explicitly excluding studies that only examined physical or built environmental factors (e.g., air pollution; greenspace) [2]. While we recognize the interaction of social and physical environments and the use of physical environments as a proxy for social environments, narrowing our scope allowed us to focus on how the more ambiguous concept of social environments is operationalized and the potential pathways that begin with a strictly social exposure (e.g., stress). At the same time, this definition and the social-ecological framework allows for the inclusion of a broad variety of exposures that range from self-reported interpersonal dynamics (e.g., levels of social support) to policy-level factors (e.g., state welfare programs). Studies identified for inclusion by title and abstract, or for which inclusion criteria could not be determined based on title and abstract alone, underwent full-text review and data extraction by KG and RP. Lastly, we conducted citation review of the systematic reviews identified by our search to identify any additional original studies that met our inclusion criteria, which underwent full-text review and data extraction by KG and RP.

### 2.3. Data Extraction and Analysis

For each included study, data were extracted on (1) study design; (2) population; (3) definition and operationalization of social environment measures; (4) cognitive/dementia outcome(s); (5) primary analytic approach; and (6) mechanistic pathways by which social environments are theorized to shape the cognitive aging and dementia outcome. We used a three-tiered social ecological model as a framework to organize our findings [4,5]. This model recognizes that individuals are embedded in varying levels of environments that interact with each other and the individual to shape health behaviors and produce health outcomes (Figure 1A). Studies that met inclusion criteria were categorized as investigating an interpersonal, community, and/or policy level social environment. We defined interpersonal level social environments as those measures that focus on person-to-person relationships and emphasize the participation or interaction of the individual. Community level social environments encompass neighborhood or community-level dynamics (e.g., neighborhood demographics) that exist outside of the interpersonal sphere but may influence it. Policy level social environments were defined as structural factors, typically determined by policies or systems put into place by decision-makers that produce and shape community or interpersonal level social environments (e.g., redlining and similar policies, which produced racially/ethnically segregated neighborhoods that persist today [15]).

We noted when studies fell into more than one of these social ecological levels. We then thematically coded each group of studies based on the measurements of its social environment exposure(s) to identify trends in the existing body of literature and opportunities for future investigations [14].

## 3. Results

Structured searches identified 7802 non-duplicate studies published from 1968 to 2020. Of these, 133 were included in full-text review and 114—comprised of 108 original studies and 6 systematic reviews—met criteria for data extraction (Table 2 and Table 3). The 6 identified systematic reviews were published between 2004 and 2020 (Table 2). Four of these reviews aggregated studies of one or more dimensions of interpersonal-level social environments, specifically including social network measures, social support, social activity participation and living alone [16,17,18,19]. Two systematic reviews aggregated studies of community-level social environments (neighborhood socioeconomic measures, social disorder/crime and community social support) [20,21]. From the 6 systematic reviews, an additional 15 studies met inclusion criteria, contributing to a final sample of 123 original studies (Table 3). Figure 2 provides the CONSORT flow diagram for the number of identified, excluded, and included studies.

### 3.1. Original Studies

Original studies, published between 1989 and 2020, presented findings for populations from North America (n = 50), Europe (n = 44), Asia (n = 25), Brazil (n = 1), Nigeria (n = 1), Australia (n = 1) and the World Health Organization’s Study on global AGEing and adult health (SAGE), which is comprised of samples from China, Ghana, India, the Russian Federation, and South Africa (n = 1). These studies had a mean sample size of 6931 (range: 89–184,633).

In the identified studies, 72 examined cognitive function and cognitive change, 37 examined cognitive impairment and 25 investigated dementia. Methods of assessment varied across studies, ranging from in-depth neuropsychological test batteries to short assessment tools developed for research (e.g., Telephone Interview for Cognitive Status) or clinical assessment (e.g., Mini Mental Status Examination (MMSE)). Dementia studies relied on clinical criteria established in the Diagnostic and Statistical Manual (DSM) version in use at the time of the study or International Classification of Disease (ICD) diagnostic codes pulled from electronic health records.

When sorted into the 3-tiered social-ecological model (Figure 1A), 91 examined interpersonal level social environments [22,23,24,25,26,27,28,29,30,31,32,33,34,35,36,37,38,39,40,41,42,43,44,45,46,47,48,49,50,51,52,53,54,55,56,57,58,59,60,61,62,63,64,65,66,67,68,69,70,71,72,73,74,75,76,77,78,79,80,81,82,83,84,85,86,87,88,89,90,91,92,93,94,95,96,97,98,99,100,101,102,103,104,105,106,107,108,109,110,111,112], 37 examined community-level social environments [9,50,68,80,113,114,115,116,117,118,119,120,121,122,123,124,125,126,127,128,129,130,131,132,133,134,135,136,137,138,139,140,141,142,143] and 3 examined policy-level social environments [67,109,110] (Table 2, Figure 1B). Of these, 4 examined both interpersonal and community level environments [50,68,72,80], 2 examined interpersonal and policy level environments [67,109], and 1 examined social environments at all three levels of the social ecological model [110].

Fifteen studies examined prospectively or retrospectively collected time-varying exposures: social support [106]; social strain [93]; formal and informal social activity participation [32,42,51,56,67,83,84,87,96,107,143] social network structure (e.g., number of family/friends) [32,79,83]; participation in national health insurance [67]; having a confidante at earlier ages [102]; and income inequality [114].

### 3.2. Interpersonal Level Social Environments and Cognitive Aging and Dementia

Using thematic coding, four major categories of interpersonal level social environment exposures emerged: (1) perceived levels of social support/strain (n = 40); (2) formal social activity participation (n = 39); (3) informal social interaction/isolation (n = 47); and (4) social network structure (n = 25). Social support/strain was derived through asking participants about perceived access to specific types of emotional or financial support, if they had someone they could trust or call upon for help, and/or reciprocity in their relationships. Studies examining formal social activity participation asked participants about the frequency with which they attended public meetings or religious services or participated in club or society activities. By contrast, studies of informal participation examined frequency of in-person visits or phone calls from friends and family or the number of close personal contacts who live nearby. Social network structure was assessed by network size (e.g., the number of friends/family who were regularly in contact), network composition (e.g., more family members than friends) or network complexity (e.g., number of different types of relationships/roles in their network). Twenty studies used one or more validated measures. Among these, the Lubben Social Network Scale [144] was the most popular, used in 8 studies. Another 5 validated measures [145,146,147,148,149] were each used in 2 studies and 9 validated measures [150,151,152,153,154,155,156,157,158] were each used in 1 study. Seventy-five (82%) studies found stronger social environments were associated with better cognition and lower dementia risk, while 8 studies (9%) found strong social environments to be associated with poorer cognition and higher dementia risk. Sixteen studies (18%) found no significant associations.

### 3.3. Community Level Social Environments and Cognitive Aging and Dementia

Studies of community level social environment exposures for cognitive aging and dementia fell into two operational groups: those that used demographic data for geographically defined areas (e.g., census tract; n = 27), and those that relied on participant responses to define the community level social environment (n = 10; Table 3). The most common demographic data measure was neighborhood socioeconomic status, using either single indicators (e.g., median income; n = 8), or indices of area deprivation (n = 19). Other demographic data measures included income inequality calculated at the U.S. state or metropolitan statistical area [114,125]; racial/ethnic residential composition [116]; and the density of older adults living alone [120]. Six studies aggregated study participant responses to define area-level social activity participation [80] and social support/cohesion [50,68,123,124,143]. Five studies used perceptions of neighborhood social cohesion (e.g., trust; friendliness with neighbors) [50,126,133,134,141]. Twenty-nine studies (78%) found neighborhood environments that were safer, had higher levels of trust and/or higher socioeconomic status were associated with better cognition or lower dementia risk. One study found having more neighborhood social ties was associated with higher risk of cognitive impairment; 11 studies (30%) observed no associations.

### 3.4. Policy Level Social Environments and Cognitive Aging and Dementia

Our search identified 3 studies that included a social environmental measure at the policy level [67,109,110]. All 3 investigated the policy-level environment in conjunction with an interpersonal environmental measure. Specifically, Andel, et al., (2012) examined dementia odds among Swedish twins resulting from work-related chronic stress exposure, a measure that incorporated social support (interpersonal) and the balance of one’s employment demands versus control to assess job strain (policy) [109]. Chiao et al., (2019) examined several time-varying predictors of late-life cognitive function, including social participation and volunteering (interpersonal) and the implementation of the National Health Insurance program in Taiwan (policy) [67]. Luo et al., (2019) examined social environment measures from all three social ecological levels with cognitive status: social activity participation (interpersonal); neighborhood SES (community); and the availability of employment services and old age income subsidies (policy) in China [110]. All 3 studies observed significant associations between enriched social environments and better cognition or lower dementia risk.

### 3.5. Pathways between Social Environment Exposures and Cognitive Aging and Dementia Outcomes

Nearly all studies (93%) proposed one or more theoretical pathways by which social environments could influence cognitive aging and dementia outcomes. These included increasing cognitive stimulation and cognitive reserve, reducing psychosocial stress, and influencing health behaviors. Interpersonal level social environment pathways were best articulated in the study by Ertel, et al., (2008; p. 1220)

“Social integration may reduce the onset of (vascular) conditions and help to ameliorate their consequences through direct neurohormonal pathways and behavioral modifications. Social ties may create pressure, either through explicit reminders or implicit behavioral norms, to take care of oneself, for example, by careful management of chronic conditions. Another possible mechanism is through cognitive aspects of social interactions: by presenting complex cognitive and memory challenges, social interactions may enhance cognitive reserve, improve compensation in response to neurophysiologic decline, and increase resilience after neuronal injury. Finally, contacts with friends and loved ones may provide a greater sense of purpose and emotional validation that has direct neurohormonal benefits” [25].

Similarly, in studies of community level social environments, low community deprivation and high community social cohesion were theorized to impact cognitive aging and dementia by increasing social and material resources that promote good health behaviors, reduce stress, and otherwise minimize cardiovascular risk factors. Additionally, enriched community level social environments may promote cognitive stimulation as well as facilitate increased social engagement and positive affect. Along these lines, policy level environments that increased access to resources (e.g., National Health Insurance; income subsidies) were theorized to minimize stress and increase opportunities for social engagement and complex social interactions.

## 4. Discussion

The aim of this scoping review was to better understand how social environments have been operationalized in epidemiologic studies of cognitive aging and dementia. We initiated this review with a broad definition of social environments set forth by Yen and Syme (1999) that informed our search criteria and inclusion and exclusion processes [2]. We then used a social ecological framework to organize our findings, and thematic coding to further disentangle the different social environmental factors under study [4,5]. We identified 91 studies that examined interpersonal level social environment exposures that measured perceived social support/strain, formal social activity participation, informal social interaction/isolation, and social network structure. Thirty-seven studies examined community level social environment exposures using geocoded census-type measures or aggregated place-based survey responses to measure area-based socioeconomic status/deprivation, community cohesion or community engagement. Three studies examined policy level social environments, which included national health insurance, government income subsidies, employment services and job strain. Seven of these studies examined social environment measures that fell into more than one social ecological level. Most studies theorized social environments to effect cognitive aging and dementia outcomes through shaping access to material and social resources, psychosocial stress and health behaviors, including cognitively enriching activities. Of 123 identified studies, 107 found one or more enriched social environment exposures were associated with better cognitive aging outcomes, while 27 found no association for at least one social environment exposure examined. Nine studies observed a negative association between one or more enriched social environment exposures and cognitive aging outcomes. In many of these studies, negative associations may be due to reverse causation (e.g., higher social support needed among those with worse cognition), and were typically limited to sub-populations (e.g., women, but not men) or distinct components of a measure (e.g., social networks comprised of more family than friends).

Our findings helped to map the current literature and indicate a substantial foundation of scientific research into social environments and cognitive aging and dementia outcomes from which to build—especially at the interpersonal-level where we identified the largest number of studies. Systematic reviews and meta-analyses could help to determine the quality of studies identified in this scoping review and provide greater insights into the strength and consistency of reported associations.

We also observed conceptual overlap in many of the measures used at the interpersonal level, yet substantial variability in operationalization. Several validated measures of social support are available to assess multiple aspects of the construct and that accommodate cultural and linguistic diversity [159]. Increased use of these existing measures, where possible, will increase direct comparability of studies.

### 4.1. Advancing the Study of Social Environments and Cognitive Aging and Dementia

The findings from this scoping review highlight two key opportunities to advance our understanding for how the social environment shapes cognitive aging and dementia and, ultimately, improve public health efforts that reduce dementia risk and disparities. One is the advancement of transdisciplinary research that allows for broader theorization and contextualization of how social environments get into the brain to impact cognitive aging and dementia outcomes. A second opportunity is found in technological and methodological advances that allow for more sophisticated investigation of social environmental exposures. We discuss each of these in turn.

The social ecological model provided a useful tool for organizing the social environment exposures identified in this review [4,5]. Importantly, the social ecological perspective recognizes the potential role of multiple interacting social ecological levels for individual health and is explicit about the potential impact of intervening on a given risk factor: the higher the ecological level the larger—albeit potentially less specific—the impact on health. What is often missing from this perspective is the biological mechanism. Theoretically broadening this model to incorporate a more explicit biopsychosocial perspective will enhance understandings for *how* social environments get into the brain to shape cognitive aging and dementia outcomes and clarify which opportunities for intervention will best balance time, cost, and precision. One emerging field in which to center this work is the study of the “exposome” [160,161]. The exposome, conceptualized by cancer epidemiologist Christopher Paul Wild, is comprised of all potential exposures that contribute to chronic disease and is set in juxtaposition to the genome [160]. Wild organizes the exposome into three broad domains: internal (e.g., hormones, metabolites), specific external (e.g., environmental pollutants; health behaviors), and general external (e.g., socioeconomic status; climate) [161]. The recent interest in advancing the science of the exposome provides an innovative opportunity for social environment researchers to engage in a transdisciplinary conversation about the specific linkages between external exposures and biological processes [162]. At the same time, exposome research will benefit from the theoretical and methodological contributions of social scientists and community intervention researchers who can help to disentangle the causal chains and interactional processes between social environments, physical environments and other external exposures that fall within the broad external categories set forth by Wild. Incorporating existing frameworks, such as the Social Ecological Model or the Built Environment Change framework [163], will help to clearly articulate the mechanistic pathways amenable to intervention.

Methodological and technological developments also provide an opportunity to advance the field. For example, more ubiquitous data access from video, cell phones, and internet technology [164], as well as advances in biomarkers for dementia pathologies [165] create newfound opportunities to investigate potential relationships between social ecological exposures and the biological hallmarks of dementia. Advances in analytic methods for complex data structures and causal inference methods also increase the capacity for investigating these questions. One example is with formal social network analysis (SNA), which uses graph theory to map individuals and the ties between them, and to study the pattern of relationships within a given network (e.g., centrality of the individual in a network; how all network actors are connected) [166,167,168,169]. Through the use of SNA, researchers could define how risk factors for dementia are distributed within social networks [169]; advance understanding of risk by investigating the relational structure between symptoms, co-occurring disease and dementia biomarkers [170]; or evaluate the impact of community-based interventions [171,172].

### 4.2. Limitations

We acknowledge several limitations to this review. First, we theoretically distinguished between social environments and physical environments in our search criteria, excluding terms for the latter. We acknowledge that some measures of the physical environment (e.g., rural/urban, population density) may also be used in research as *proxies* for a social environment, or that social environments may provide contextual meaning to physical environmental measures (e.g., poor air quality due to cook stoves versus industrial development). These complexities were not fully accounted for in this review. However, our search did allow for the inclusion of studies examining a wide variety of social environmental factors even if the authors did not use the term “social environments” to frame their research. Studies identified by our search that examined both social environments and physical environments were not excluded, though data on physical environments were not extracted. Relatedly, we did not incorporate specific language into our search terms to account for all possible policy-level social environments, which likely contributed to our small sample of policy-level studies—all of which also incorporated social environmental measures from interpersonal or community levels. Nonetheless, this limitation in our approach starkly highlights how few studies have empirically examined multiple ecological levels of the social environment in the same study—a central component of the social ecological model. Finally, our scoping review search criteria were not designed to capture every observational study in this topical area. Rather, we emphasized longitudinal observational studies as these designs provide the best opportunity for integrating social environmental theory with causal inference in the field. This approach likely restricted our ability to exhaustively identify all epidemiologic studies in this area.

## 5. Conclusions

We conclude by highlighting that few studies considered the interactional nature of multiple levels of our social ecology, especially with regards to policy-level exposures. We also emphasize that new opportunities for transdisciplinary dialogue and the use of advanced and novel methodological approaches will help to move the field forward through enhancing our understanding of how social environments may contribute to disparate cognitive and brain health outcomes. A more robust body of science, as proposed, will also help public health researchers and practitioners to define and implement the policies and interventions most likely to reduce the risk of cognitive aging and dementia.

## Figures and Tables

**Figure 1 ijerph-18-07166-f001:**
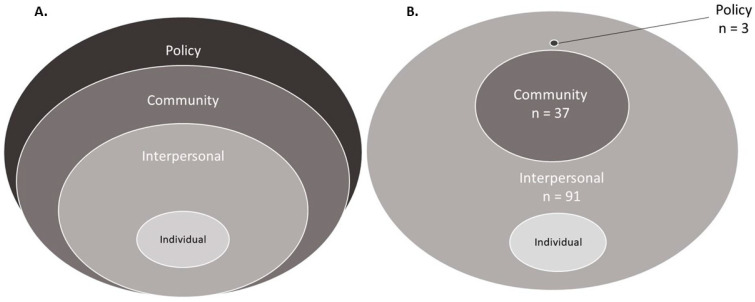
(**A**) Conceptual 3-tier social ecological model for individual cognitive aging and dementia outcomes used to organize studies of social environments included in this review. (**B**) The 3-tier social ecological model rescaled to reflect the proportion of single studies that met inclusion criteria in each ecological level.

**Figure 2 ijerph-18-07166-f002:**
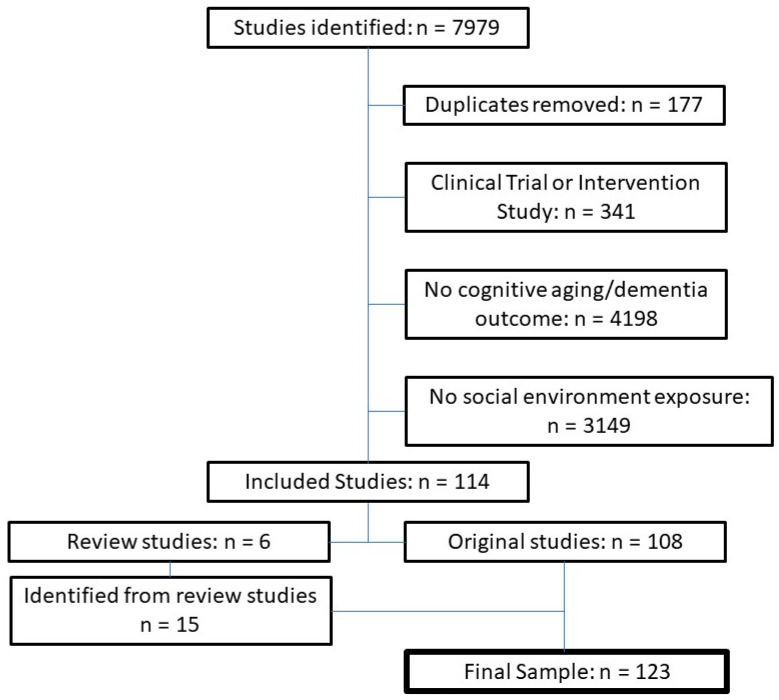
CONSORT flow diagram of study exclusion and inclusion.

**Table 1 ijerph-18-07166-t001:** Search criteria and terms implemented in the PubMed and Web of Science databases from inception through October 2020, limited to English language peer-reviewed studies.

Key Inclusion Criteria	Search Terms Applied in PubMed
Cognitive aging and dementia outcome	Cognition[mesh] OR “Cognition Disorders”[mesh] OR cognitive function OR episodic memory OR executive function OR working memory OR “mini-mental state examination” OR aging[mesh] OR Alzheimer disease[mesh] OR dementia
	AND
Social environment exposure	(“social capital” OR “social class” OR social environment OR socioeconomic factors OR income inequality OR “neighborhood characteristics” OR “social context” OR “social milieu” OR norm OR culture OR “social integration” OR “social network”)
	AND
Observational/epidemiologic study	“surveys and questionnaires”[mesh] OR “linear models”[mesh] OR “logistic models”[mesh] OR “longitudinal studies”[mesh] OR “follow-up studies”[mesh]

**Table 2 ijerph-18-07166-t002:** Social environment measures and social ecological level for systematic review studies that met inclusion criteria.

Citation	Sample of Studies	Review Study Objective	Measure(s) of Social Environment	Key Social Environment Finding
Desai et al., 2020 [16]	12	Review and meta-analyze longitudinal studies on living alone and incident dementia.	Living alone	Living alone was associated with 1.3 times the risk (95 % CI: 1.13–1.51) of incident dementia.
Fratiglioni et al., 2004 [17]	7	Review evidence for longitudinal effects of social network, physical leisure, and non-physical activity on cognition and dementia.	Varied by study and included quantitative (e.g., network ties) and qualitative (e.g., social support) markers.	Poor social network characteristics were associated with higher risk of cognitive decline or lower cognitive performance in 5 of 7 studies and dementia risk in 3 of 6 studies.
Kuiper et al., 2015 [18]	19	Review and meta-analyze the association between social relationship aspects (e.g., social network size, social participation, loneliness) and incident dementia in the general population	Six categories: social network size, participation in group activities, social contact frequency, loneliness, social network satisfaction (e.g., having good relations with others), other (e.g., perception of reciprocity)	Risk of dementia was higher among those with low social participation (RR: 1.41 (95% CI: 1.13–1.75)) and less frequent social contact (RR: 1.57 (95% CI: 1.32–1.85)).
Penninkilampi et al., 2018 [19]	33	Review and meta-analyze the evidence of association between social engagement, loneliness, and dementia risk from observational studies	Three categories: poor social engagement, good social engagement or loneliness.	Poor social engagement was associated with increased dementia risk (RR = 1.41, 95% CI 1.21–1.65).
Besser et al., 2017 [20]	22	Review evidence of association between neighborhood built and social environments and cognition in older adults.	Four categories: SES (e.g., income), demographics (e.g., race/ethnicity), social disorder (e.g., crime)/social climate/social ties (e.g., social support)	Evidence was moderately strong for neighborhood SES, moderate for neighborhood demographics and weak for psychosocial disorder.
Wu Y-T et al., 2015 [21]	15	Review evidence of association between community environment and cognitive function in later life	Community-level socioeconomic status/deprivation	Eleven of 15 studies found significant associations between community-level socioeconomic status/deprivation and late-life cognition.

**Table 3 ijerph-18-07166-t003:** Social environment level, cognitive aging and dementia outcome and study design for included individual studies.

Citation	Interpersonal	Community	Policy	Cognitive Impairment	Dementia	Cognitive Function	Longitudinal	Cross-Sectional	Time-Varying Exposure
Cadar et al., 2018 [9]		E			x		x		
Amieva et al., 2010 [22]	AB				x		x		
Ayotte et al. 2013 [23]	A					x		x	
Ellwardt et al., 2015 [24]	B					x	x		
Ertel et al., 2008 [25]	D					x	x		
Evans et al., 2018 [26]	AD					x	x		
Fan et al., 2015 [27]	CD				x			x	
Fankhauser et al., 2015 [28]	CD				x		x		
Fankhauser et al., 2017 [29]	AD			x			x		
Fratiglioni et al., 2000 [30]	D				x		x		
Ge et al., 2017 [31]	A					x		x	
Glei et al., [32]	BCD					x	x		x
González-Moneo et al., 2016 [33]	A			x				x	
Bae et al., 2018 [34]	D			x				x	
Gow et al., 2016 [35]	AD					x	x		
Grande et al., 2018 [36]	D				x		x		
Green et al., 2008 [37]	ABD					x	x		
Griffin et al., 2020 [38]	D					x	x		
Gureje et al., 2011 [39]	CD				x		x		
Haslam et al., 2014 [40]	CD					x	x		
Heser et al., 2014 [41]	AC				x		x		
Hikichi et al., 2017 [42]	ACD					x	x		x
Holtzman et al., 2004 [43]	AD					x	x		
Hosking et al., 2017 [44]	CD					x	x		
Bennett et a., 2006 [45]	BD					x	x		
Hwang et al., 2018 [46]	C					x	x		
James et al., 2011 [47]	C					x	x		
Jedrziewski et al., 2014 [48]	CD			x		x	x		
Kats et al., 2016 [49]	AD					x	x		
Jiang et al., 2020 [50]	C	GH				x		x	
Kim, C et al., 2016 [51]	CD				x		x		x
Kim, YB et al., 2019 [52]	BCD					x	x		
Kotwal et al., 2016 [53]	ABC			x	x			x	
Krueger et al., 2009 [54]	ABC					x		x	
Kuiper et al., 2020 [55]	BD					x	x		
Biddle et al., 2019 [56]	CD					x	x		x
Lee et al., 2020 [57]	BC					x		x	
Liao et al., 2017 [58]	A					x	x		
Litwin et al., 2016 [59]	BC					x		x	
Ma et al., 2018 [60]	D				x		x		
Malek Rivan et al., 2019 [61]	A			x		x		x	
Marioni et al., 2015 [62]	ABC				x	x	x		
Marseglia et al., 2019 [63]	ABCD				x		x		
Mattavelli et al., 2016 [64]	D				x		x		
McHugh et al., 2017 [65]	AD					x	x		
Murata et al., 2019 [66]	ACD				x		x		
Chiao, 2019 [67]	C		J			x	x		x
Murayama, Miyamae et al., 2019 [68]	A	GH		x				x	
Murayama et al., 2013 [69]	B					x	x		
Noguchi et al., 2019 [70]	A					x	x		
O’Shea et al., 2018 [71]	CD					x		x	
Ouvrard et al., 2017 [72]	D	F			x		x		
Paúl et al., 2010 [73]	AD			x				x	
Pillemer et al., 2016 [74]	A					x		x	
Pugh et al., 2020 [75]	AC					x	x		
Rafnsson et al., 2020 [76]	D				x		x		
Rodriguez et a., 2018 [77]	B				x	x	x		
Conroy et al., 2010 [78]	A					x		x	
Röhr et al., 2020 [79]	AD				x	x	x		x
Sakamoto et al., 2017 [80]	C	G				x		x	
Santini et al., 2017 [81]	CD			x			x		
Seeman et al., 2001 [82]	AB					x	x		
Sharifian, Kraal et al., 2020 [83]	D					x	x		x
Sommerlad et al., 2019 [84]	CD				x	x	x		x
Sörman et al., 2015 [85]	D				x		x		
Stoykova et al., 2011 [86]	ABC					x	x		
Thomas et al., 2011 [87]	CD					x	x		x
Tomioka et al., 2016 [88]	C					x	x		
Crooks et al., 2008 [89]	AD				x		x		
Wang, C et al., 2016 [90]	B			x				x	
Wang, H-X, 2002 [91]	C				x		x		
Wang, Z et al., 2020 [92]	ABC				x		x		
Wilson et al., 2015 [93]	A			x			x		x
Wu, J et al. 2020 [94]	B				x		x		
Zunzunegui et al., 2003 [95]	CD					x	x		
Aartsen et al., 2002 * [96]	C					x	x		x
Bassuk et al., 1999 * [97]	AD			x			x		
Andrew & Rockwood, 2010 * [98]	ABC					x	x		
Blasko et al., 2014 * [99]	BC				x		x		
Deng et al., 2019 [100]	C			x				x	
Brown et al., 2009 * [101]	AB					x		x	
Camozzato et al., 2015 * [102]	AB				x		x		x
Holwerda et al., 2014 * [103]	D				x		x		
Fabrigoule et al., 1995 * [104]	C				x		x		
Magaziner & Cadigan, 1989 * [105]	BD					x		x	
Riddle et al., 2015 * [106]	AB			x	x		x		x
Saczynski et al., 2006 * [107]	D				x		x		
Salinas et al., 2017 * [108]	AB				x		x		
Andel et al., 2012 [109]	A		I		x			x	
Luo et al., 2019 [110]	C	F	K			x	x		
Dyer et al., 2020 [111]	AD					x	x		
Gow et al., 2013 [112]	AD					x		x	
Hunter et al., 2018 [113]		F				x	x		
Kim, D et al., 2016 [114]		E				x	x		x
Kim, GH et al., 2017 [115]		E		x			x		
Kovalchik et al., 2015 [116]		F				x	x		
Lang et al., 2008 [117]		F				x		x	
Letellier et al., 2018 [118]		F			x		x		
Letellier et al., 2020 [119]		F				x		x	
Liu et al., 2019 [120]		E			x		x		
McCann et al., 2018 [121]		F		x				x	
Meyer et al., 2018 [122]		F		x	x		x		
Murayama et al., 2018 [123]		GH		x				x	
Murayama, Ura, et al. 2019 [124]		G		x				x	
Peterson et al., 2019 [125]		E				x		x	
Sharifian, Spivey et al., 2020 [126]		H				x	x		
Shih et al., 2011 [127]		F				x		x	
Sheffield et al. 2009 [128]		F		x		x	x		
Wight et al., 2006 [129]		E				x		x	
Wörn et al., 2017 [130]		E				x	x		
Wu, Y-T et al., 2015 [131]		F		x				x	
Zeki et al., 2011 [132]		F				x	x		
Zaheed et al., 2019 [133]		H				x		x	
Zhang et al., 2019 [134]		H				x		x	
Basta et al., 2008 * [135]		F		x				x	
Deeg et al., 2005 * [136]		F		x			x		
Lee et al., 2011 * [137]		F				x		x	
Boardman et al., 2012 [138]		F				x	x		
Clarke et al., 2012 [139]		F				x		x	
Danielewicz et al., 2016 [140]		E		x				x	
Estrella et al., 2020 [141]		H				x		x	
Fernández-Blázquez et al., 2020 [142]		F		x	x		x		
Hikichi et al., 2020 [143]		G				x	x		x
Total	91	37	3	25	37	72	84	39	15

* Identified from a review study. A. Social support/strain B. Social network structure C. Formal social activity participation D. Informal social interaction E. Community demographic (single indicator) F. Community demographic (index of multiple indicators) G. Aggregated participant demographics H. Participant perceptions of community I. Work site policy/structure J. National Health Insurance Participation K. Employment services; income subsidies.

## Data Availability

Not applicable.

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
