# Peer review of "Operationalizing Social Environments in Cognitive Aging and Dementia Research: A Scoping Review"

_ijerph, 2021, doi:10.3390/ijerph18137166_

Round 1

Reviewer 1 Report

I would like to congratulate the authors for a very well written manuscript in an important field of research. It is really impressive! I only have a few, minor comments before acceptance: 

Results: 

Please check table 3 for typos (e.g. for the reference Sheffield, 2009). 

Line 178, what does 633 stand for?

Line 298, page number is missing in the reference

Discussion:

On lines 315-330 the authors provide a summary of the Results. I find this too lengthy and repetitious and suggest that this part is condensed a bit. 

Lines 342-345: I don't know whether there is a bias but I wonder why the authors say that "only xxx used the Lubben Social Network scale". I can't see the reason for (potentially) favouring the Lubben scale over another scale. Or is it not meant to, if so the authors may just revise the sentence. 

Line 411: studies mentioned twice

Reviewer 2 Report

Dear Authors, 

I read you work and it is very interesting and well documented.  It is nice that you performed a scoping review, but i would suggest you to run a systematic review using PRISMA. I make this recommendation since you have a good amount of reference.

Thank you! 

Reviewer 3 Report

This study is a scoping review, aiming to define the definition of social environment in cognitive aging and dementia research.

Major issue:

  1. It is somewhat difficult to disentangle the physical environment from social environment. Take the air populations for example; although the air pollution is defined as physical attribute, the underlying issues of air pollution might result from poverty and environment inequality in the undeveloped and/or developing countries. it is important to distinguish between the ‘survival’ emissions of the poorest – for example, their use of polluting cookstoves which cause severe health damage – and the ‘rich’ emissions generated by rich and powerful elites to maintain their lifestyles.
  2. Redundant paragraphs were presented from line 72 to 76. Also, it is unnecessary to point out the difference between scoping reviews and systematic reviews in the 1st section of Method
  3. According to the guidelines of systematic scoping review (Peters et al., 2015), The review objective(s) and specific review question(s) must be clearly stated. However, I didn’t see the research question clearly.
  4. Although the authors intended to use 3-tiered social ecological perspective (i.e., “interpersonal, community, and polity”) to portrait the definition of social environment and its impact on cognitive aging, it seems previous literature reviews (Table 2) have already documented the author’s ambitious for carrying out this scoping review. The authors might need to provide more strong rationale for this scoping review.  
  5. Redundant paragraphs and tables. There was several repeated information in the results sections and tables. Please check.

Round 2

Reviewer 2 Report

Dear Authors, 

Thank you for your reply in my review. My suggestion to your team is to have a final English check.

Author Response

We thank the reviewer for providing a close read of our manuscript. Below we have detailed the changes made in this revision to correct grammatical errors and/or improve readability.

Lines 49-52: We reworded a run-on sentence into three sentences as follows:

In other words, social environments are not physically or geographically contained. Rather, they are formed by various layers of interpersonal relationships, the cultural and demographic characteristics of our communities and societal power dynamics. These layers social policies, opportunities, resources, and norms that ultimately impact the distribution of health and disease [3].

Lines 126-128: We removed the use of a semi-colon and joined two statements to smooth the readability of the following sentence:

Lastly, we conducted citation review of the systematic reviews identified by our search to identify any additional original studies that met our inclusion criteria, which underwent full-text review and data extraction by KG and RP.

Lines 131-133: We reworded the 6th data extraction criteria for clarity, as follows:

6) mechanistic pathways by which social environments are theorized to shape the cognitive aging and dementia outcome.

Lines 154-156: We rephrased the statement of final study counts for clarity, as follows:

Of these, 133 were included in full-text review and 114 – comprised of 108 original studies and 6 systematic reviews – met criteria for data extraction (Tables 2, 3).

Lines 172-173: We revised the sentence as follows:

Original studies, published between 1989 and 2020, presented findings for populations from…

Line 196: We removed an extra “and”

Line 210: We revised the statement as follows:

“e.g., the number of friends/family who were regularly in contact”

Line 235-236: We added “having” to the following sentence:

One study found having more neighborhood social ties was associated with higher risk of cognitive impairment; 11 studies (30%) observed no associations.

Line 243: We replaced “and” with “versus” as follows:

…and the balance of one’s employment demands versus control to assess job strain (policy).

Line 244: We removed the colon and added “including” to the following sentence:

Chiao et al., (2019) examined several time-varying predictors of late-life cognitive function, including social participation and volunteering (interpersonal) and the implementation of the National Health Insurance program in Taiwan (policy) [68]

Line 273: We replaced “high-resourced and positive” community environments with “enriched”

Line 282: We added an “and” where needed

Line 303: We corrected the typo, changing “for” to “or”

Lines 384-386: We split a lengthy sentence into two as follows:

Finally, our scoping review search criteria were not designed to capture every observational study in this topical area. Rather, we emphasized longitudinal observational studies as these designs provide the best opportunity for integrating social environmental theory with causal inference in the field.

Reviewer 3 Report

The authors have revised their manuscript quite well. 

Author Response

Thank you.